## Barriers and facilitators to patient uptake and utilisation of digital interventions for the self-management of low back pain: a systematic review of qualitative studies

Malene Jagd Svendsen [1,2] Karen Wood Wood,[3] John Kyle,[3] Kay Cooper,[4] Charlotte Diana Nørregaard Rasmussen,[2] Louise Fleng Sandal [1] Mette Jensen Stochkendahl [1,5] Frances S Mair,[3] Barbara I Nicholl[3]

► Prepublication history and additional materials for this paper is available online. To view these files, please visit the journal online (http://dx.doi.org/10.1136/bmjopen-2020-038800).

For numbered affiliations see end of article.

**Correspondence to**
Malene Jagd Svendsen;
mas@nrcwe.dk

## ABSTRACT

**Objectives** Low back pain (LBP) is a leading contributor to disability globally. Self-management is a core component of LBP management. We aimed to synthesise published qualitative literature concerning digital health interventions (DHIs) to support LBP self-management to: (1) determine engagement strategies, (2) identify barriers and facilitators affecting patient uptake/utilisation and (3) develop a preliminary conceptual model of barriers and facilitators to uptake/utilisation.

**Design** Systematic review following PRISMA (Preferred Reporting Items for Systematic Reviews and Meta-Analyses) guidelines.

**Data sources** MEDLINE, Embase, CINAHL, PsycINFO, Cochrane Library, DoPHER, TRoPHI, Web of Science and OT Seeker, from January 2000 to December 2018, using the concepts: LBP, DHI and self-management.

**Eligibility criteria** Peer-reviewed qualitative study (or component) examining engagement with, or barriers and/or facilitators to the uptake/utilisation of an interactive DHI for self-management of LBP in adults (community, primary or secondary care settings).

**Data extraction and synthesis** Standardised data extraction form was completed. COREQ (Consolidated criteria for Reporting Qualitative research) checklist was used to assess methodology. Data was synthesised narratively for engagement strategies, thematically for barriers/facilitators to uptake/utilisation and normalisation process theory was applied to produce a conceptual model.

**Results** We identified 14 191 citations, of which 105 full-text articles were screened, and five full-text articles from four studies included. These were from community and primary care contexts in Europe and the USA, and involved 56 adults with LBP and 19 healthcare professionals. There was a lack of consideration on how to sustain engagement with DHIs. Examination of barriers and facilitators for uptake/utilisation identified four major themes: IT (information technology) usability–accessibility; quality–quantity of content; tailoring–personalisation; and motivation–support. These themes informed the development of a preliminary conceptual model for uptake/utilisation of a DHI for LBP self-management.

**Conclusions** We highlight key barriers and facilitators that should be considered when designing DHIs for

### Strengths and limitations of this study

► This systematic review of qualitative studies explored barriers and facilitators for the uptake and utilisation of digital health interventions for low back pain (LBP) to inform the future design and implementation processes of such interventions.

► Searches in multiple databases and independent data extraction, quality appraisal and detailed data analysis are strengths of our review. However, our search strategy revealed that literature in the field of digital self-management for LBP is sparse as only a small number of eligible studies were identified.

► Given the limited literature, it is possible that not all important barriers and facilitators for uptake and utilisation have been identified and thus our conceptual model must be considered preliminary.

LBP self-management. Our findings are in keeping with reviews of DHIs for other long-term conditions, implying these findings may not be condition specific.

**Systematic review registration** A protocol for this systematic review was registered with https://www.crd.york.ac.uk/PROSPERO/ (CRD42016051182) on 10 November 2016. https://www.crd.york.ac.uk/PROSPERO/display_record.php?ID=CRD42016051182

## BACKGROUND

Low back pain (LBP) affects approximately 12% of the general population at any point in time;[1] it is the leading contributor to disability worldwide[2] and is associated with significant personal[3] and societal costs.[4 5] Self-management approaches are consistently recommended in clinical guidelines as a core component of LBP management;[6 7] however, adherence to self-management strategies has proved challenging, especially without support and reinforcement.[8 9] Digital health interventions (DHIs), health interventions accessed through a computer, mobile phone or other

handheld device, involving a web-based programme, desktop programme or application; offer a potential method of supporting self-management,[10–12] and particularly the possibility of tailoring self-management advice, may hold significant potential for people with LBP.[13] DHIs or 'digital therapeutics' are becoming increasingly popular and, as technological innovations increase, it is expected that this trend will continue.[14 15] Until now, two systematic reviews have examined the use of DHIs to support the self-management of LBP. The first, by Garg *et al*, aimed to determine which web-based interventions are of benefit to patients.[16] They identified nine randomised controlled trials (RCTs), including a total of 1796 participants. Four trials studied online cognitive behavioural therapy (CBT) with the remaining five trials studying web-based interventions with interactive features such as a virtual gym, testimonials or moderated discussion groups. Garg *et al* reported that online CBT approaches appeared to reduce catastrophising and improve patient attitudes, while studies of web-based interventions with interactive features used a variety of diverse outcome measures yielding inconclusive results; thus, making it difficult to draw firm conclusions regarding long-term impact for people with LBP.

The second review, by Nicholl *et al*, aimed to appraise the evidence concerning the use of interactive DHIs to support patient self-management of LBP with a focus on the outcome measures used and reported effects.[17] They identified six completed RCTs studying digital tools for the self-management of LBP including a total of 2706 participants. Nicholl *et al* reported that only one of the six completed RCTs observed a between-group difference in favour of the digital intervention, with none of the studies demonstrating any evidence of harm. The authors noted that there was considerable variation in the nature and delivery of the interventions and inconsistency in the choice of outcomes and concluded that the current evidence base for DHIs to support the self-management of LBP remained weak.

Yet, hundreds of smartphone applications (apps) related to LBP are currently available on the app market, most developed with very little scientific rigour.[18] In order to facilitate the development of appropriate and effective self-management DHIs for those with LBP, it is important to have an understanding of the factors that help or hinder user engagement and adherence. Across different conditions, multiple barriers and facilitators to engaging with DHIs have previously been identified, including issues such as motivation and support, digital literacy, privacy, usability, quality and tailoring.[17 19] However, given the diverse range of DHIs available, it can be difficult to apply these findings to a specific patient population or piece of technology. Understanding the experience of users of DHIs designed specifically to assist self-management of LBP would help determine how to optimise DHIs for this group of users.

The purpose of this systematic review was therefore to synthesise and critically appraise the published qualitative literature concerning the use of DHIs to promote self-management of LBP in order to address the following two research questions:

1. What engagement strategies at the time of enrolment have been used in DHIs aimed at supporting patient self-management of LBP?
2. What are the barriers and facilitators to patient uptake and utilisation of digital interventions to support self-management of LBP?

The final objective of the systematic review was to develop a preliminary conceptual model of barriers and facilitators to uptake and utilisation of digital interventions to support self-management of LBP.

## METHODS

### Protocol and registration

This review was registered in the International Prospective Register of Systematic Reviews, PROSPERO, registration no. CRD42016051182[20] and reporting is consistent with the Preferred Reporting Items for Systematic Reviews and Meta-Analyses (PRISMA) statement.[21]

### Patient and public involvement

This research was done without patient involvement. Patients were not invited to comment on the study design and were not consulted to develop patient-relevant outcomes or interpret the results. Patients were not invited to contribute to the writing or editing of this document for readability or accuracy.

### Eligibility criteria

Qualitative studies that examine engagement, barriers and/or facilitators to patient uptake and utilisation of digital interventions for the self-management of LBP were included; inclusion and exclusion criteria are outlined in table 1.

### Information sources and search strategy

A systematic search of bibliographic databases (MEDLINE, Embase, CINAHL, PsycINFO, Cochrane Library, DoPHER, TRoPHI, Web of Science and OT Seeker) was conducted after the search strategy had been developed in collaboration with a librarian at the Norwegian University of Science and Technology and experienced researchers in the field of LBP and digital health interventions. The search strategy has previously been described and published by Nicholl *et al*.[17] Reference and citation tracking was used to identify relevant references. All databases were searched for publications using three groups of concepts: (1) low back pain, (2) digital intervention and (3) self-management. The search was conducted in three waves using the same search strategy: the first for publications added between January 2000 and March 2016, then a subsequent updated search for articles added between March 2016 and October 2016, and lastly, articles added between October 2016 and December 2018. Limitation of year of publication from 2000 onwards was chosen as

**Table 1** Inclusion and exclusion criteria

| Inclusion criteria | |
| --- | --- |
| Study type | ▶ Published in peer-reviewed journals between 1st January 2000 and 18th December 2018. |
| | ▶ Original qualitative studies, studies involving secondary qualitative analysis of qualitative data and qualitative studies that were part of a mixed methods study (provided the qualitative methodology was described). |
| | ▶ Qualitative data collected via questionnaires or other methods not involving direct contact or observation of participants were eligible for inclusion provided questions were answered using free text and analysed using a qualitative approach. |
| | ▶ Qualitative data describing barriers and/or facilitators to the uptake or utilisation of digital interventions or containing a description of an engagement strategy (ie, any method used to get people to enrol into the study) from a patient or HCP's perspective. |
| Language | ▶ Published in English, Danish or Norwegian. |
| Participants | ▶ Adults >18 years with LBP or HCPs providing care for such patients. |
| Setting | ▶ Community, primary or secondary care and other specialist contexts including those that recruit via media. |
| Digital intervention | ▶ Any intervention accessed through a computer, mobile phone or other handheld device, involving a web-based programme, desktop programme or application that provided self-management content (consistent with previous reviews[17 45]). |
| | ▶ Interventions must involve an element of interaction between the user and the digital interface; this was defined as information being taken from users which then provided some form of automated feedback and/or advice in response. |
| | ▶ Interventions that included face-to-face contact were only included if this interaction was in addition to an automated, interactive digital component without direct HCP mediation. |
| Exclusion criteria | |
| Study type | ▶ Descriptive case studies, lexical studies that analyse natural language data presented as qualitative results, literature or systematic reviews, meta-analyses, studies without a sampling procedure (ie, no clear description of recruitment strategy) and commentary articles written to convey opinion or stimulate discussion with no research component. |

HCP, healthcare professional; ;LBP, low back pain.

our review was aimed at understanding current experiences of digital health technologies, justified by emerging Internet access around the millennium and the developing field of DHIs that followed, and further supported by other systematic reviews of digital interventions.[16 22 23]

The complete search strategy, including specifications on the use of title, keywords or abstract screening is documented in online supplemental file 1.

### Study selection
All identified citations were uploaded to DistillerSR software (Evidence Partners, Ottawa, Canada) and duplicates were removed. Title and abstract screening were performed by two of four independent reviewers (JK, MJS, KC and KWW) using DistillerSR. Any disagreement between the two reviewers at title screening level resulted in inclusion of the citation to abstract level and subsequently any disagreement at abstract level resulted in inclusion of the citation to the full-text screening level. Full-text screening was also performed by two of four independent reviewers (JK, MJS, KC and KWW) with any discrepancies at this level being resolved through discussion mediated by a third party (BIN, CDNR, MS and KC).

### Data extraction
A comprehensive, standardised data extraction template designed specifically for this review in DistillerSR was used by two of four independent researchers (JK, MJS, BIN and KWW). Where available, information collected included the study title, authors, citation, year of study and publication, country, inclusion/exclusion criteria, aim, setting, characteristics of the digital intervention, recruitment methods, method of qualitative data collection and analysis, participant numbers and characteristics, any engagement strategies, barriers or facilitators identified either by the authors or in participant quotes, conclusions, limitations, funding sources and any potential conflicts of interest declared.

### Quality appraisal
The complete 32-item Consolidated criteria for Reporting Qualitative research (COREQ) checklist[24 25] was used to assess the methodological quality of the articles progressing to data extraction. Two of three reviewers (BIN, KC and KWW) independently identified whether each of the 32 items were reported or not, and descriptive information was provided where possible. Disagreements between reviewers were resolved through discussion. A priori cut-off points were not determined as studies were not excluded on the basis of methodological quality due to a lack of clear agreement on how best to appraise qualitative literature.[26] Two of the included articles report on the qualitative evaluation of the same intervention but were treated as separate articles for quality appraisal.[27 28]

### Data synthesis and analysis
Information on the engagement strategies, defined as methods used to recruit and initially motivate participants to enrol in the DHI study, in each study was described narratively as this was only provided descriptively in the included studies. Our data synthesis of barriers and facilitators to patient uptake and utilisation of the DHI for LBP involved a thematic approach.[29] Data on barriers and facilitators were extracted from results and discussion

sections of the included studies. Each item of extracted data was initially coded by one reviewer (MJS). When new codes appeared during the analysis of a particular article, the articles that had previously been examined were re-read and re-coded if appropriate. This continuous adjustment was carried out in cooperation with a second reviewer (KWW). Emergence and mapping of codes were discussed at coding clinics to ensure construction of themes that were internally homogeneous and externally heterogeneous (ie, no data excluded due to lack of a suitable theme, and no data falling between two themes or fitting into more than one theme)[30][31] (MJS, KWW, FM and BIN). This resulted in a coding taxonomy for mapping identified codes as barriers or facilitators for each theme.

A preliminary conceptual model of barriers and facilitators to uptake and utilisation of DHIs to support self-management of LBP was developed by mapping the identified themes to the four constructs of Normalisation Process Theory (NPT). NPT is a sociological theory developed to explore the process of implementing a new complex intervention, in this case it can help explain how people individually and collectively embed DHIs into everyday practice.[32][33] The identified themes were mapped to NPT constructs by four reviewers (KWW, FM, BIN and JK) using the coding framework presented in table 2. This approach has been successfully applied in other systematic reviews of DHIs for chronic disease self-management issues[19][34][35] and provides a solid conceptual basis from which to understand barriers and facilitators to patient and HCP uptake and utilisation of DHIs. Any themes that could not be coded to the NPT constructs were carefully noted to ensure that themes outside the scope of NPT would still be captured to assure appropriateness of the model.

## RESULTS

### Study selection

Of 14191 citations identified, 5973 were excluded as duplicates; 8113 were excluded following title and abstract screening (7436 at title level and 677 at abstract level) and a further 100 citations were excluded after full-text screening. Overall, five full-text articles were included in the review (figure 1). These articles described four separate studies and included a total of 75 participants. The two articles[27][28] reporting on the same study (Oneself) consisted of a qualitative evaluation of a website[28] and a mixed-method reporting of the same qualitative data combined with quantitative (pre-use and post-use surveys and log files) data.[27] As these two studies included the same qualitative data and user quotes, they were combined for analysis purposes.

### Study characteristics

The Get Well Fast[36] and Oneself studies[27][28] were undertaken between 2006 and 2008 in the Netherlands and Switzerland, respectively. The MyBehaviorCBP study was

**Table 2** Core constructs of Normalisation Process Theory (NPT)[32][33] and related coding framework for development of preliminary conceptual model of barriers and facilitators to uptake and utilisations of digital interventions to support self-management of LBP

| Core constructs of NPT | Coding framework |
|---|---|
| Coherence (sense making work; enrolling with the DHI): development of an individual and collective understanding of the new intervention when faced with operationalising it. | ► How people understand and view the benefits versus disbenefits of DHIs and decide whether it is appropriate for them to use.<br>► Motivation and willingness to commit to self-management activities. |
| Cognitive participation (engagement work; engaging with the DHI): relational work to build and sustain engagement with a new intervention. | ► Willingness to 'buy into' the DHI and whether it is a legitimate means to promote self-management of LBP.<br>► Issues relating to the support provided to use the DHI and level of engagement of HCPs involved with the DHI. |
| Collective action (operationalisation work; utilising the DHI): investment of effort and resources to enact the new intervention. | ► Ease of use, accessibility and appropriateness of the DHI.<br>Resources, training, workload and technical support.<br>► Perceived quality and trustworthiness of DHI content and function. |
| Reflexive monitoring (appraisal work; maintaining engagement with DHI): evaluation of the impact of the new intervention on individuals and groups along with any reconfigurations suggested. | ► How people judge the new DHI and the self-monitoring work that accompanied uptake of the DHI.<br>► Ability to tailor to an individual's needs. |
| Codes falling outside the NPT framework | |
| | ► Inherent personal attributes such as personal physical or cognitive abilities that could promote or inhibit DHI use. |

DHI, digital health intervention; HCP, healthcare professional; LBP, low back pain.

conducted in the USA between 2012 and 2014,[37] while the study period for the Swedish Web-BCPA study was not reported.[38] The characteristics of the study participants are summarised in table 3. No information was reported on comorbidities or ethnicity and only limited information on participant socioeconomic status was included.

DHI delivery mode varied between studies. In the Oneself, Get Well Fast and Web-BCPA studies, the DHI

**PRISMA 2009 Flow Diagram**

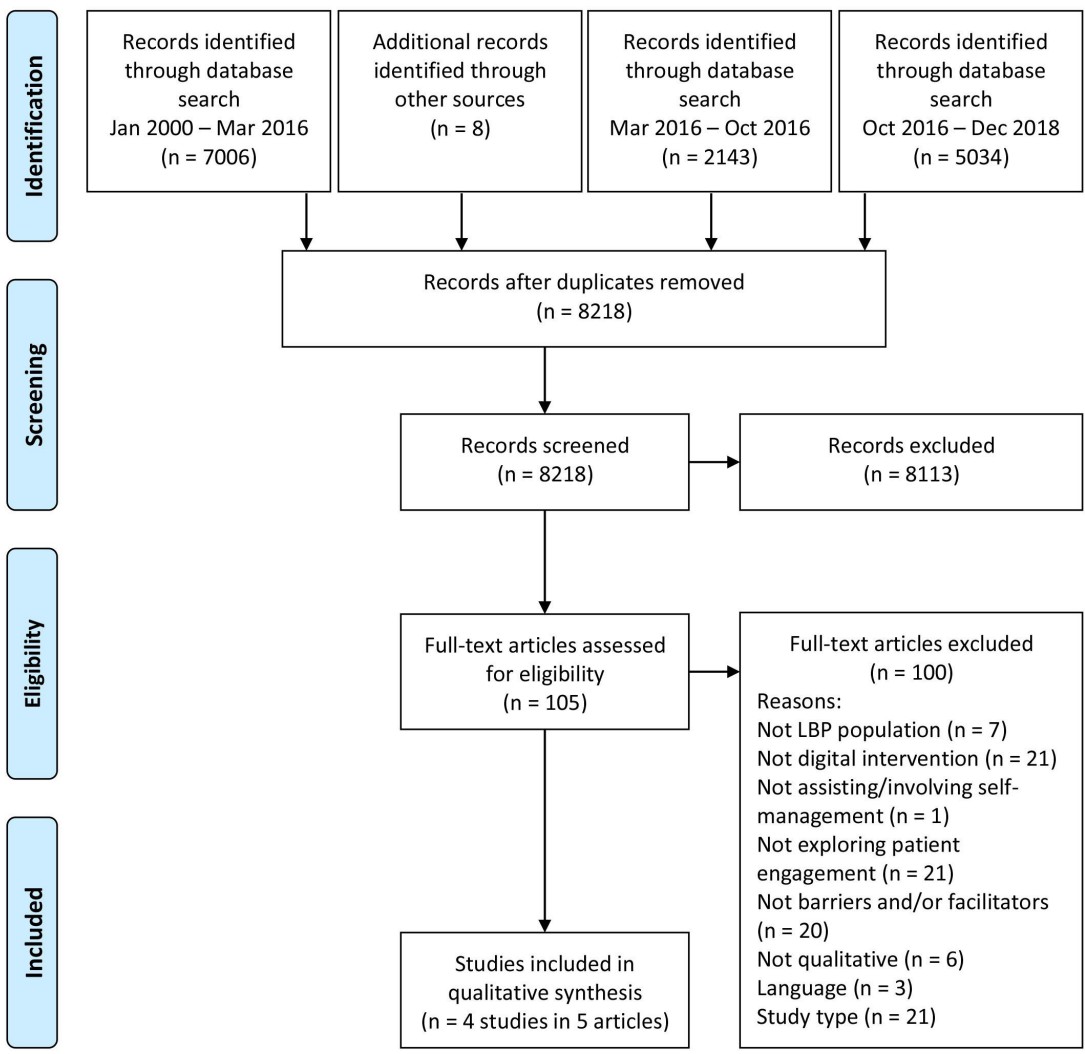

**Figure 1** PRISMA flow diagram illustrating the screening process (Adapted from Moher et al[21]). LBP, low back pain; PRISMA, Preferred Reporting Items for Systematic Reviews and Meta-Analyses.

consisted of information available on websites to which participants had either open access[27 28] or had personal log-ins.[36 38] The content of the MyBehaviorCBP intervention was delivered to participants via a mobile phone app.[37] Two of the studies tailored the content of their DHI to the individual participant by collecting information about the users and providing content that matched their needs;[36 37] in the Get Well Fast study, content was tailored-based on patient reports on pain, limitations, treatment, counselling, reintegration to work, work situation and work characteristics, relations at work, personality and daily activities,[36] while the MyBehaviorCBP intervention collected sensory data from the users' smartphone (accelerometer signals and geolocation) and patient self-reported physical activity logs.[37] Three interventions offered time limited programmes of either 5 weeks[36 37] or

8 weeks,[38] while the fourth intervention was an open-to-access website with no time restrictions[27 28] (table 4).

### Qualitative components of included studies

Sampling procedures used for the qualitative component of the included studies (table 4) were described for three of the studies as an invitation to participants to take part in an interview.[27 28 36] Several sampling strategies were used, including purposive[27 28] and convenience sampling,[27 28 36] while in another study participants were sampled consecutively.[38] In the further study, where the qualitative component was part of a self-administered survey, all participants took part.[37] Qualitative interviews were conducted via telephone,[36] in the participant's home[27 28 38] or at a local university,[27 28] healthcare centre[38] or council building.[38] All of the interviews were semi-structured, recorded and

**Table 3** Participant characteristics of included studies

| Study; country | Year of study | Number of participants in qualitative study | Age range | Sex (%) | SES |
|---|---|---|---|---|---|
| Oneself Switzerland[27 28] | 2006–2008 | n=18 | 28–72 years<br><29 years: n=1<br>30–39 years: n=3<br>40–49 years: n=5<br>50–59 years: n=6<br>>60 years: n=3 | 50% female | Education: Secondary school: n=2; high school or equivalent: n=11; university degree: n=5 |
| Get Well Fast Netherlands[36] | 2008 | n=28<br>OP+=11<br><br>OP−=8<br>Employee: 9 | 40–50 years | OP: N/R<br>Employee: 33% female | White and blue-collar workers. Various levels of education |
| MyBehaviorCBP USA[37] | 2012–2014 | n=10 | 31–60 years | 70% female | N/R |
| Web-BCPA Sweden[38] | N/R | n=19 | 27–60 years | 79% female | Education: Elementary school: n=2; secondary school: n=12; university degree: n=5)<br>Employment: Permanent employment: n=12; temporary employment: n=3; unemployed: n=3; social benefits: n=1 |

DHI, digital health intervention; N, number; N/R, not reported; OP+, occupational physicians who recruited patients into DHI; OP–, occupational physicians who did not recruit patients into DHI; SES, socioeconomic status.

either transcribed verbatim[27 28 38] or as written descriptions of answers including quotes.[36] For the MyBehaviorCBP study,[37] free-text answers from the electronic exit survey were extracted. Data was then analysed inductively,[27] using grounded theory,[28] thematically[36 37] and using content analysis[38] to identify common themes. Just one article[28] referred to data collection and analysis continuing until data saturation was achieved.

## Quality appraisal

The comprehensiveness of reporting varied across the included studies (online supplemental file 2) and ranged from 12 (38%) to 21 (67%) of the 32-item COREQ checklist.[27 38] Items within domain 1 (Research team and reflexivity) generally had very poor reporting with several items not reported by any studies, for example, researcher occupation and experience and training were not reported by any of the included studies. All studies reported sampling procedure, sample size, setting of data collection, description of sample, recording, derivation of themes, quotations presented, consistency of data and findings and clarity of major themes

## Engagement strategies

We defined engagement strategies as any method used to recruit and initially motivate participants to enrol in the DHI study. The identified engagement strategies included: use of mailing lists of retired personnel;[37] mailing list for a university wellness centre;[37] or invitation from occupational physician (OP) or healthcare professional (HCP).[27 28 36] In addition, the Oneself study advertised for participation through media: radio (project leader and managers interviewed about project at local radio station), television (rheumatologists involved in project spoke about project on local television station) and through a press conference for which the major daily journals from the area were invited.[27 28]

## Barriers and facilitators for uptake and utilisation of digital health interventions

We identified four major themes: (1) IT (information-technology) usability and accessibility, (2) quality and amount of content, (3) tailoring and personalisation and (4) motivation and support (table 5). Under each theme, both barriers and facilitators were identified. Distinction between uptake (initial engagement) and utilisation (use) in the included studies was not possible, and they are therefore treated as one. Participant quotes are provided in the text to substantiate the data for each theme. More exemplar quotations are provided in online supplemental file 3.

### IT usability and accessibility

The first theme that emerged concerned functionality and usability, IT affinity or access and convenience of the DHI. A flexible and convenient structure with high user-friendliness aided use of DHIs.[36 38] Inclusion of a variety of

**Table 4** Participant inclusion criteria, sampling procedure for qualitative component and characteristics of digital intervention in included studies

| Study | Inclusion criteria for digital health intervention | Inclusion criteria and sampling procedures for qualitative study | Characteristics of digital health intervention |
|---|---|---|---|
| Oneself[27 28] | ► Anyone could register and use the Oneself website. | ► Registered users of Oneself for at least 6 months.<br>► Visited the website at least three times.<br>► Suffering from chronic LBP (duration not defined).<br>► Living in the Italian part of Switzerland.<br>► Purposive and convenience sampling<br>► Invitation to participate in interview sent via email to eligible users.<br>► Reminder email sent after 2 weeks to anyone who had not responded.<br>► 238 users invited to participate, 18 agreed. | ► Open access website containing:<br>► Library - textual educational information on back pain.<br>► Radio - 10×2 min recorded audio messages on relevant topics.<br>► Gym - videos demonstrating stretching, stabilisation and mobilisation exercises accompanied by photographs and written descriptions.<br>► Forum - users could interact with other users and HCPs, monitored by a content manager.<br>► Chat room - users could interact with other users and HCPs. Once a week, a HCP would be available to discuss specific topics selected from conversations published on the Forum.<br>► Specialist answers - information on topics suggested by users.<br>► Testimonials - users could share stories and comment on other users' stories.<br>► Ability for users to request information they felt lacked on the website. |
| Get Well Fast[36] | ► Employees of KLM Royal Dutch Airlines or National Railways and their OPs.<br><br>Employee criteria:<br>► Contracted for at least 12 hours per week.<br>► Absent from work for a minimum of 2 weeks due to non-specific back or neck pain.<br>► No serious health problems defined as 'warning flags: for example, fever, pain in arms or legs, serious disease'.<br>► Ability to speak and write in Dutch.<br>► Internet access. | ► Users of the Get Well Fast website.<br>► The employees' OPs.<br><br>► All employees using the website and OPs were invited to participate in an interview.<br>► Convenience sample | ► Web-based, 5 weeks programme during which the employee completed four questionnaires and received tailored information via a personal digital diary.<br>► Based on weekly questionnaires, information about advice on improving physical fitness, setting a daily timetable, pain-coping strategies and exercise instructions is provided.<br>► Employees spent around 15 min/day reading information, completing questionnaires and following exercises.<br>► Employee's OP had access to the employee's diary and received reports when the employee completed a questionnaire, detailing the employee's condition, current treatments and absence details. |

| Study | Inclusion criteria for digital health intervention | Inclusion criteria and sampling procedures for qualitative study | Characteristics of digital health intervention |
|---|---|---|---|
| MyBehaviorCBP[37] | ▶ Aged 18–65 years<br><br>▶ History of chronic back pain (≥6 months).<br><br>▶ Willingness to use MyBehaviorCBP app on an Android mobile phone (own or provided by study).<br><br>▶ Reasonable level of outdoor movement (eg, travelling to and from work).<br><br>▶ Not being significantly housebound.<br><br>▶ Fluent in English<br><br>▶ Basic level of mobile proficiency. | ▶ All participants received web-based exit survey; one question was open ended and results from this component of the study are included in this review. | ▶ 5-week app based programme during which participants received recommendations for PA.<br><br>▶ App tracks participant's mobility state and geolocation using in-phone sensors or manual input. Recurring patterns of PA form base for new PA recommendations.<br><br>▶ Week 1 - baseline period: no recommendations were given.<br><br>▶ Week 2 and 3 - control phase: PA recommendations were random, generic and unrelated to participants' past behaviour.<br><br>▶ Week 4 and 5 - experimental phase: PA recommendations generated by MyBehaviorCBP based on PA behaviour during control phase.<br><br>▶ Participants were blinded to when the different PA recommendation forms were activated. Participants completed a daily in-phone survey regarding ease of following recommendations, how many recommendations they followed and their emotional state. |
| Web-BCPA[38] | ▶ Aged 18–63 years.<br><br>▶ Persistent musculoskeletal pain with duration of at least 3 months in the back, neck, shoulder and/or generalised pain.<br><br>▶ OMPSQ score ≥90, screening for psychosocial factors that indicates an estimated risk for long-lasting pain and future disability.[46]<br><br>▶ Work ability of at least 25% (assessment method N/R).<br><br>▶ Familiar with written and spoken Swedish.<br><br>▶ Internet and computer access. | ▶ Participants must have spent at least 15 min per module in five of eight modules.<br><br>▶ Participants had to have reached their 4-month follow-up assessment<br><br>▶ Participants contacted consecutively with information about interview study in conjunction with 4-month follow-up.<br><br>▶ Formal invitation subsequently via telephone. | ▶ Website-based Web Behavior Change Program for Activity (Web-BCPA) in combination with MMR.<br><br>▶ Web-BCPA consisted of eight modules: (1) pain, (2) activity, (3) behaviour, (4) stress and thoughts, (5) sleep and negative thoughts, (6) communication and self-esteem, (7) solutions and (8) maintenance and progress.<br><br>▶ Modules contained information, assignments and exercises delivered as educational texts, videos and writing tasks.<br><br>▶ Participants could access one new module/week during the first 8 weeks of rehabilitation, and had access to the website 24/7 for 4 months. |

app, application; HCP, healthcare professional; LBP, low back pain; MMR, multimodal rehabilitation; N/R, not reported; OMPSQ, Örebro Musculoskeletal Pain Screening Questionnaire; OP, occupational physician; PA, physical activity.

**Table 5** Factors affecting uptake and utilisation of DHIs for self-management of LBP

| Theme | Subtheme | Barriers | Facilitators |
|---|---|---|---|
| IT usability and accessibility | Functionality and usability | ► Too much choice between functions | ► Flexible structure and navigation |
| | | ► Fixed advancement pace | ► Conveniently arranged |
| | | ► Issues logging into DHI | ► Variation of media types (text, audio and video) |
| | | ► *Low user-friendliness | ► Reminders and notifications |
| | | ► *Issues logging into DHI | ► High user-friendliness |
| | | ► *Low level of functionality (eg, registration, navigation, help desk) | ► *High user-friendliness |
| | IT affinity | ► Lack of affinity with computers | ► Enjoying working with a computer |
| | | ► *Lack of affinity with web-based programmes | |
| | Access and convenience | ► Not able to choose starting time of DHI | ► Easily accessible with low effort |
| | | ► *No access to computer during consultation | ► Accessible at all hours and locations |
| | | | ► Accessible even during periods with severe pain symptoms |
| | | | ► Ability to take all the time needed |
| Quality and amount of content | Quality of content | ► Contradictory content between DHI and HCP | ► Trustworthy content and source |
| | | | ► Easily understandable content |
| | | | ► High quality of content |
| | | | ► Steady content |
| | | | ► *Appropriate content |
| | Amount of content | ► Too much content to choose from | ► A lot of content to choose from |
| | | ► Too much information to fully comprehend | |
| Tailoring and personalisation | Tailoring, specificity and personalisation | ► Content not tailored to individual needs and/or pain severity | ► Content accounting for individual needs and/or pain severity |
| | | ► Content perceived not new or relevant | ► Self-identification in content |
| | | | ► Opportunity to influence treatment |
| Motivation and support | Personal attributes and resources | ► Adhering to biomedical model of LBP | ► High level of awareness and self-management of LBP |
| | | ► Seeing LBP as a marginal problem | ► Aware that LBP would not be fixed with a medical solution and ready to accept active role |
| | | ► Preferring other treatment regimens, for example, with human contact | ► Emotional and cognitive resources, for example, motivation, interest, commitment and self-confidence in self-management of LBP |
| | | ► Lack of knowledge about LBP and treatments | ► Enjoy solution focussed work |
| | | ► Physical health (eg, pain, fatigue) | |
| | | ► Psychological symptoms | |
| | Support to use DHI | ► HCP unsupportive of use of DHI | ► HCP supportive of use of DHI |
| | | ► No support from authorities | ► Support from family |
| | | | ► Support from authorities |
| | | | ► Support from other suffers (eg, successful testimonials) |

Continued

| Table 5 | Continued | | |
|---|---|---|---|
| Theme | Subtheme | Barriers | Facilitators |
| | Features of DHI | ▶ DHI not guiding or supporting participants enough (eg, to plan for execution of physical activity recommendation from DHI) | ▶ Interaction/interactivity<br>▶ Information about self-management of LBP<br>▶ Goal-setting<br>▶ Action-planning<br>▶ Follow-up and evaluation<br>▶ Adjusting treatment related to setbacks and progress<br>▶ Monitoring own progress in graphs<br>▶ Variation of content<br>▶ Update of content |
| | HCP factors for support of patients | ▶ *Time restrictions of consultations<br>▶ *Difficulty keeping DHI in mind during consultations<br>▶ *Difficulty providing patients with accurate information about DHI<br>▶ *Perceiving no benefit of DHI compared with usual treatment<br>▶ *Preferring other treatment regimens, for example, with human contact | ▶ *DHI a good medium for counselling employees |

*Occupational physician perspective.
DHI, digital health intervention; HCP, healthcare professional; IT, information technology; LBP, low back pain.

media types such as video was also appreciated[27 28] as well as getting reminders or notifications from the DHI.[27 28]

'Usually I went on the website when I read the newsletter. I read the letter and then I'm there, it's like a conditioned reflex (Woman, 49, nurse)'.[27 28]

On the other hand, low user-friendliness and problems with logging in were barriers for use of DHIs for both study participants and HCPs.[36] A fixed starting point or set advancement pace were also demotivating for some users.[38] Affinity with computers and web-based programmes highly affected uptake of DHIs. Participants with a high level of computer affinity and who enjoyed working on a computer expressed positive feelings towards using DHIs,[38] whereas lack of computer affinity was an important barrier for uptake of the intervention.[36] Accessibility to a computer was surprisingly not a requirement for uptake to the study. When computers were readily available, DHIs were considered easy to access with unlimited 24 hours access.[27 28 38]

'… thanks to the programme (the Web-BCPA) I was able to perform the basic body awareness exercises of my own choice… and to repeat those that I felt most effective as many times that I preferred… the flexibility made it mine (the rehabilitation) (Woman, participant)'.[38]

Even during periods with severe pain symptoms, a DHI was considered an attainable and effortless option as participants did not have to go anywhere (eg, a healthcare centre).[27 28 38]

### Quality and amount of content
Quality and amount of content provided in DHIs affected use for both participants and HCPs. Trustworthiness of the source and information provided facilitated use, and participants seemed to be reassured when knowing the content had been reviewed and validated by HCPs.[27 28 38] For participants, richness and consistency of content facilitated use,[27 28] especially when the content was easily understandable.[36]

'Knowing that there is a serious website where there are contributions, it strengthens you a bit (Woman, 37, teacher)'.[28]

Likewise, content that suited the patients was appreciated by HCPs.[36] On the other hand, when participants experienced contradictory advice from their HCP and the DHI, this was a barrier for using the DHI.[36] Large volumes of information or too much content to choose from also limited uptake and utilisation, particularly in relation to the amount of time required to go through it.[27 28 36]

'There is a lot of information, probably almost too much, don't you think? (Man, 47, bank director)'.[27 28]

## Tailoring and personalisation

The participants' perception of the degree of tailoring and personalisation of the content to their needs was the third major theme affecting use of DHIs for self-management of LBP. Self-identification increased utilisation of DHIs when participants were able to recognise themselves in the content, for example, in the information and explanations about pain and symptoms, or thoughts related to dealing with LBP.[27 28 38]

'It gives you descriptions and you say: this stuff here… I see it, I see it! I recognise myself in it, I recognise myself here (Man, 58, teacher)'.[27 28]

When the content of the DHI accounted for the individual participant's activities, needs or pain severity it further encouraged use of the DHI.[36–38]

'I really liked the personalisation. I thought it was a nice touch. Suggestions were more specific and tailored, which for me made them more relevant and likely for me to use them (Participant)'.[37]

Participants appreciated the opportunity to influence their own rehabilitation by being able to select exactly what they wanted from a variety of options that fitted their situation.[37 38]

'Previously I had read about CBT (cognitive behavioural therapy), but I had never thought of it as a help for my condition… I want to compare this rehabilitation with a smorgasbord from which is it easy to taste (Participant)'.[38]

When content was not tailored to the individual participant or the participant's pain severity, it was experienced as a barrier for use of the DHI as it was not perceived to apply to their situation. This in turn would negatively impact the participant's motivation and sustained engagement.[28 36] Content that was not perceived relevant or new to the participant could also lead to a feeling of hopelessness as participants' got the impression that there was no solution to their problem.[28]

## Motivation and support

The fourth major theme related to the participants motivation and support, and included subthemes related to the personal attributes and resources of participants, support to use DHIs, features of DHIs and lastly HCPs' perceptions and how they affect HCPs' support of DHIs. Specific participant attributes impacted the utilisation of DHIs; already being involved or being ready to accept an active role in rehabilitation,[27] and having motivation, interest, commitment and confidence in self-managing LBP facilitated use.[27 28 38] Enjoying solution focussed work, for example, as experience from day job, was also a facilitator.[38] Contrary, not wanting to take an active role,[27] or preferring other treatment regimens[27] hindered use, as

well as lacking information about treatments[38] or preferring other available treatment regimens, for example, with human contact.[36] Relying on a HCP to find a solution[27 28] or seeing LBP as only a marginal problem, led to lower motivation for use of the DHI.[27] Furthermore, use of DHIs was constrained by physical[36 38] or psychological[38] restrictions. Getting support from a variety of sources facilitated use; both support from outside and within the DHI. Support from family, authorities and HCPs was perceived as encouraging,[38] and so were successful testimonials from other users whose LBP symptoms had improved.[27 28]

'When you are going through a moment when you have backache and you read a testimony which says 'yes, there is someone who was able to do it', it gives you hope (Woman, 28, academic researcher)'.[27 28]

Not having HCPs or local agencies (eg, authorities) support in their use of the DHI held participants back from using DHIs to manage their LBP.[36 38]

'I expected more commitment from my OP (occupational physician) (Employee)'.[36]

Features of DHIs could both facilitate and restrain use. DHIs that were interactive, used goal-setting and action-planning and had a great variation of content encouraged use.[37 38] Participants also appreciated information that guided them on how to self-manage their LBP (eg, exercises and advice),[27 28 36–38] and some participants felt updates of content facilitated their use further.[27 28] Furthermore, DHIs that allowed participants to monitor and reflect on their own progress, improvement or goal attainment, for example, through interactive graphs, were considered to enable self-management actions and to motivate further use.[38] Follow-up and evaluation on goal achievement was also appreciated and reinforced the importance of tailoring DHIs towards individual participant's experience.

' … days when I had a lot of pain I used to remain sedentary, and as soon as I had a better day I was eager to do all kinds of activities that day… before I started with the assignment activity planning (in the Web-BCPA) I was not aware of how my behaviour related to the days with pain, but by monitoring this over time I started to plan my daily activities in a more balanced way (Woman, participant)'.[38]

On the contrary, DHIs that did not support or guide participants enough, for example, to execute recommendations given by the DHI, were perceived as constraining.[37]

HCPs had reasons to support or not support participants' use of DHIs for self-management of LBP. HCPs either did not perceive additional benefits of DHIs compared with usual care or preferred other treatment regimens, for example, ones that involved physical contact.[36]

'The ability to touch people is an essential element in the treatment of people with back or neck pain (Occupational physician)'.[36]

HCPs also reported having too little time during consultations to support use of DHI or difficulty in keeping the DHI in mind during their consultation—and even if they remembered it, they struggled with providing patients with accurate information about the DHI.[36] However, HCPs who perceived DHIs as a good medium for counselling were positive about using and recommending DHIs.[36]

## Suggestions for improved utilisation

Participants of all included studies provided the authors with suggestions for how DHIs could be improved to facilitate continued or improved utilisation. As these items were only perceived as potential facilitators if implemented they are reported separately from the themes above. Some suggestions were improvement of usability of existing DHIs, for example, increased user-friendliness,[36] incorporation of illustrations and cartoons[36] or easier registration.[36] Optimisation of tailoring to adjust for changes over time,[36] or better adaption of physical activity recommendations that accommodated differences between weekdays and accounted for weather forecasts was also suggested.[37] System improvements that enabled the DHI to learn from participants' activity level related to their pain days was also proposed.[37] Lastly, application of a participatory approach for the process of designing DHIs was suggested.[38] Other suggestions were new features to add to DHIs, for example, direct contact to HCPs via DHI,[36] a help desk,[36] content about how to deal with LBP mentally[36] and a sophisticated reminder system with just-in-time notifications for both planning and execution of physical activities.[37]

## Developing a conceptual understanding

We applied the NPT framework (table 2) to the taxonomy of barriers and facilitators as summarised in table 5. Most of the identified codes fell within the four NPT constructs, with the exception of codes related to participants' own physical, mental and emotional health, which although affecting an individual's capacity, they are not specific actionable tasks involved in the uptake and utilisation of a DHI for LBP. Applying the NPT framework allowed us to conceptualise how the codes identified may affect the uptake and utilisation of DHIs for the self-management of LBP (figure 2), at both an individual and collective level, through the four stages of deciding whether to enrol, engage, use and maintain engagement with such a tool.

## DISCUSSION

We have conducted a systematic search of the literature to explore the methods used to encourage participation with DHIs for the self-management of LBP and the barriers and facilitators to patient uptake and utilisation of these tools. Our review identified four studies published in five articles, demonstrating that the literature remains sparse.

Our review has enabled us to develop a preliminary conceptual model for engagement and utilisation of a DHI for LBP self-management by applying the NPT framework to the barriers and facilitators identified in the included studies. The model suggests that users value DHIs that are easily understandable, which they can navigate at their own pace and which help enhance subsequent communication with HCPs, family and colleagues. Providing regular updates and prompts appears to help users engage with DHIs while the ability to interact with other users is viewed positively in terms of providing support, motivation and validation. Users expect

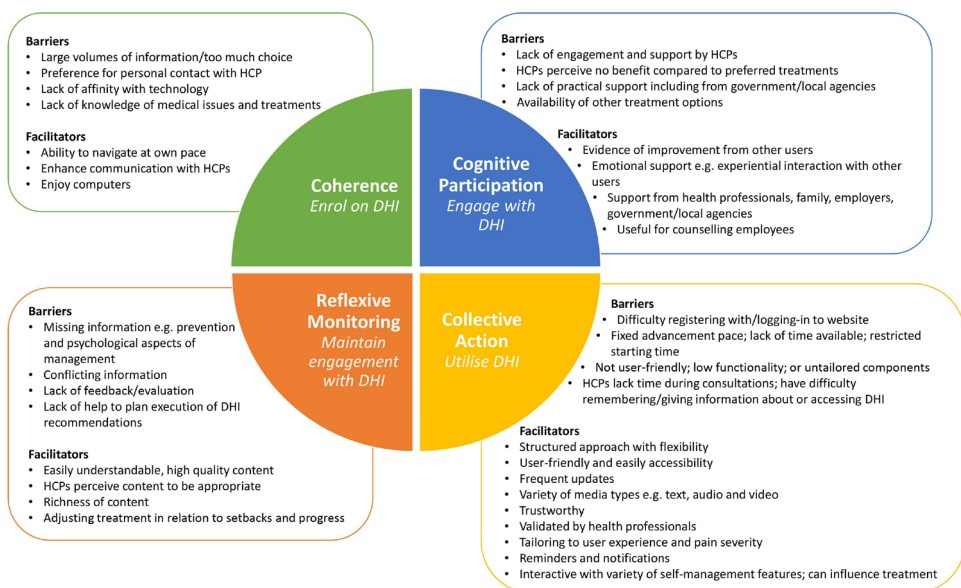

**Figure 2** Preliminary conceptual model of barriers and facilitators to uptake and utilisation of low back pain DHIs. DHI, digital health intervention; HCP, health care professional.

information to be easily accessible, structured, up-to-date and accurate, with tailoring to individual user experience being particularly valued.

Conversely, large volumes of information and lack of time appear to have a negative impact on user understanding, motivation and engagement. Lack of support or encouragement by HCPs also appears to be off putting for some while others face challenges accessing the DHIs. Participant's own attributes including the symptoms they experienced and their attitudes and preferences for treatment for LBP can further restrict capacity to self-manage and influence motivation and engagement with DHIs. Other significant barriers to user engagement and utilisation include missing or conflicting information, content that was not tailored to the individual and lack of feedback or evaluation.

In this review we explored how studies engaged participants to enrol into the study and begin using a DHI, this was mainly through identification of potential participants and subsequent invitation. Sustaining engagement beyond initial participation was not discussed in-depth in any of the included studies, some used email prompts and regular updates or newsletters. However, all studies did report participants' suggestions to improve DHIs, which mainly focussed on improving usability, (dynamic) tailoring of content, additional features to support users and the inclusion of participants in the design of DHIs. While not considered as facilitators to uptake and utilisation, some positive consequences of using the DHIs were identified by some users, for example, acquiring a vocabulary and an individual understanding of their situation, and increased confidence in self-managing their LBP, which may have reinforced users in their self-management and in turn may have increased use of DHIs. Further, some general points to increase utilisation of DHIs for LBP were highlighted by participants, including the importance of participatory involvement of patients in the development of a DHI.

### Comparison with previous literature

Although there was a significant variation in intervention recruitment and content in studies included in our review, there was a large degree of overlap in terms of the barriers and facilitators identified. Many of these are generally in keeping with the findings of other qualitative reviews for DHIs in general[19 39] as well as those looking specifically at hypertension[40] and pain management in older adults.[41] A review by O'Connor et al[19] identified four main themes relating to barriers and facilitators to engagement and recruitment to DHIs in general: personal agency and motivation; personal life and values; engagement and recruitment approach, and quality of the DHI. Another review by Hardiker and Grant[39] identified five overarching themes concerning barriers and facilitators influencing engagement with eHealth services: characteristics of users; technological issues; characteristics of eHealth services; social aspects of use; and eHealth services in use. Despite the differing terminology of the major theme headings used in these studies and those found in this review, comparison of the codes or subthemes reveals the barriers and facilitators to be broadly similar, suggesting that these may be generally transferable across DHIs. The main exception is the specific mention of security and privacy of personal information in these earlier reviews,[19 39] which was not found as a barrier in this review, although this may be due to the small number of studies in our review compared with O'Connor et al[19] and Hardiker and Grant,[39] reviews which included 19 and 50 studies, respectively.

### Functionality and general IT issues

Factors including age, ethnicity, economic status, level of educational attainment and familiarity with the Internet are recognised as being significant factors influencing access to and engagement with DHIs.[39] O'Connor et al[19] reported that a lack of digital literacy, issues accessing IT equipment or the Internet and the cost of such equipment or access are barriers to the use of DHIs. The user friendliness, design and ease of registration/logging in to a DHI were found to be significant issues for users in this review and should be carefully considered when planning a DHI.

### Quality and amount of content

Trust is a significant issue when accessing information online.[39] Clinical endorsement seems to be important to users in terms of the perceived quality of content and is in keeping with the findings of other studies in this area.[19 42] Additionally, consideration should be given to the potential for users to receive contradictory advice from the DHI and their HCP. Our findings suggest that while some users considered large volumes of information as a barrier, others valued the ability to read widely on the subject. This is thought to reflect individual preference and personal factors such as time pressures. Taking such preferences into account during the development and delivery of DHIs may increase user engagement.

### Tailoring and personalisation

It is clear from our findings that user's symptomology, prior knowledge and experience play a role in engagement. Tailoring DHIs to the user's individual symptoms and functional limitations is thought to enhance engagement[19] and may thus improve the effectiveness of the intervention. A recent review of DHIs for the self-management of LBP[17] found that no DHI for LBP used tailoring to enhance effectiveness, but commented that this could be an important means of enhancing engagement. In addition, O'Connor et al[19] recommended that any DHI should be designed and tailored to individual needs in order to reduce the self-care burden. Our findings suggest that users improved understanding of LBP and enhanced communication with their HCP during subsequent consultations. Some users commented that they would have appreciated some direct support from a HCP or that this might have enhanced engagement.

This finding is consistent with those of Steele et al[43] who during an evaluation of an Internet-based physical activity behaviour change programme, found that many participants in the Internet group would have preferred traditional face-to-face sessions. Some of the occupational physician's interviewed felt that they did not have the time and capacity within their consultation to discuss DHI use in detail.[36] If the intended purpose of a DHI is to facilitate HCP—patient communication then how the DHI or a supporting HCP dashboard could be designed to allow for efficient and useful interactions during a consultation should be considered at the design and development stage.

### Motivation and support

Personal recommendations and social support were recognised as being important in encouraging DHI user registration and in fostering engagement.[19] We found that some users valued the emotional support of being able to interact with other users. While this was a positive finding in our study and is consistent with those reported elsewhere,[39] there exists the possibility of potentially abusive or threatening behaviours developing online which could act as a barrier to some.[44] Other reports of discussion threads deviating from the original topic or containing misleading information[39] raise questions on the need for monitoring such interactive features. Our findings further suggest that an individual's personal attributes and resources (eg, emotional and cognitive) and attitudes towards self-management can influence their use of DHIs. Additional support may therefore be required for some potential users to participate and benefit from DHIs.

O'Connor et al[19] reported that some individuals do not view technology as a way of addressing healthcare needs and prefer alternative approaches to managing their health issues such as seeking support from family, friends or healthcare professionals. They also highlight the potential for DHIs to be impersonal and commented on the lack of a therapeutic relationship, particularly in situations where sensitive health or social issues are involved. Such views were also reflected among individuals, including some HCPs, in our findings. In contrast, other users appreciate the freedom to access health information at a time and place that suits the user along with the anonymity DHIs can offer,[42] issues that can be challenging for traditional healthcare services to match.

### Strengths and limitations

This systematic review was conducted by an experienced team and follows the PRISMA guidelines for the reporting of systematic reviews. Our iterative search strategy used multiple databases and involved independent data extraction, quality appraisal and data analysis by two reviewers, with a third reviewer adjudicating in the case of any disagreements.

Our review does however have some limitations. Many DHIs are developed commercially and do not undergo formal academic evaluation[15] resulting in relatively sparse literature in this area. Our search strategy involved several eligibility criteria, including that studies must be published in peer-reviewed academic journals, and as such we did not identify any grey literature. However, it is unlikely that such findings, if available, would have held scientific rigour and added to the findings of this review. Further, as our analysis and synthesis of data was based on reviewing published literature, not the original data, this could have impacted on the background context to some of the quotes used in this manuscript.

The studies included in this review[27 28 36–38] were conducted in real-life settings and as a result sampling procedures were acknowledged as being convenient, had the potential to be biassed towards individuals who found the interventions beneficial and may not have been representative of all users. Furthermore, the literature contained very limited information on user's sociodemographic characteristics. However, as a consequence of the small number of studies identified by our search strategy, we did not exclude studies on the basis of quality, potentially reducing the reliability of the findings of this review.

Finally, due to the lack of literature in this field, our conceptual model for the update and utilisation of DHIs to support the self-management of LBP is limited to four studies to date. It is possible that not all the important barriers and facilitators may have been identified, and thus our conceptual model must be considered preliminary. As more rigorous studies are conducted and reported this model should be further developed and amended. This information will be of particular use to those involved in designing and implementing DHIs focussed on self-management of LBP and more widely.

### CONCLUSIONS

Our systematic review highlights barriers and facilitators affecting the utilisation of DHIs for the self-management of LBP and identified key areas involved in embedding such interventions into everyday practice. The limited and varied quality of literature found by this review suggests that further primary research investigating the implementation of DHIs and user's experiences is required. Future research should aim to describe DHIs and their users in more detail and include descriptions of engagement strategies and barriers or facilitators encountered in order to enhance our knowledge of which approaches are likely to have the greatest impact on user engagement and outcomes, and for whom.

**Author affiliations**
[1]Department of Sports Science and Clinical Biomechanics, University of Southern Denmark, Odense, Denmark
[2]Musculoskeletal disorders and physical work demands, National Research Centre for the Working Environment, Kobenhavn, Denmark
[3]Institute of Health and Wellbeing, University of Glasgow, Glasgow, UK
[4]School of Health Sciences, Robert Gordon University, Aberdeen, UK
[5]Nordic Institute of Chiropractic and Clinical Biomechanics, Odense, Denmark

**Acknowledgements** We would like to thank our librarian adviser Ingrid Ingeborg Riphagen, Department of Public Health and Nursing, Faculty of Medicine and Health Sciences, Norwegian University of Science and Technology (NTNU).

**Contributors** Study design was developed by all authors. Title, abstract and full-text screening was performed JK, MJS, KC and KWW with any discrepancies being resolved by BIN, CDNR and MS. JK, MJS, BIN and KWW carried out data extraction. JK, KWW and MJS conducted data synthesis and analysis aided by BIN and FM. Quality appraisal was assessed by BIN, KC and KWW. LFS, CDNR and MS critically scrutinised first drafts and provided comments. All authors read, commented and approved the final manuscript.

**Funding** This project has received funding from the European Union Horizon 2020 Research and innovation programme under grant agreement No. 689040.

**Competing interests** None declared.

**Patient consent for publication** Not required.

**Provenance and peer review** Not commissioned; externally peer reviewed.

**Data availability statement** Data are available upon reasonable request.

**ORCID iDs**
Malene Jagd Svendsen http://orcid.org/0000-0001-9718-9425
Louise Fleng Sandal http://orcid.org/0000-0001-8436-1046
Mette Jensen Stochkendahl http://orcid.org/0000-0003-0297-8267

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
