## [Reviewer comments · BMJ Open]

ARTICLE DETAILS

TITLE (PROVISIONAL)	Barriers and facilitators to patient uptake and utilisation of digital interventions for the self-management of low back pain: a systematic review of qualitative studies
AUTHORS	Svendsen, Malene; Wood, Karen; Kyle, John; Cooper, K; Rasmussen, Charlotte Diana Nørregaard; Sandal, Louise; Stockendahl, Mette; Mair, Frances; Nicholl, Barbara

VERSION 1 – REVIEW

REVIEWER	Fadi Al Zoubi McGill University, Canada
REVIEW RETURNED	05-May-2020

GENERAL COMMENTS	The introduction for DHI was good. However, it is expected from the authors to define it since it involves diverse range of e-interventions. In your eligibility criteria, there is a limitation of population to include only non-specific LBP. However, the title does not reflect that, and the introduction of the manuscript does not provide any information about this type of LBP. In addition, in your search strategy you included general LBP without referring to non-specific. I would consider the definition of the DHI in the inclusion criteria in the introduction: "Any intervention accessed through a computer, mobile phone, or other handheld device, involving a web-based programme, desktop programme or application that provided self-management content". I am not familiar with the term "information specialist". Does that mean this person is a librarian or not? The authors mentioned in the study selection line 155-156 that any disagreement between the two reviewers results in inclusion of the citation to the next screening level. What is the next screening level? is it a third reviewer who decide? voting between the 4 reviewers? or full-text screening? I think there is a need to be explicit about these steps. The numbers reported in PRISMA Chart are different from the ones on results lines 208-209. The full-text screening was 100 in the text and 105 in PRISMA. The excluded studies in the text were 7436 and PRISMA 8113. Please review them As the author mentioned in the text that many of the DHIs were commercial and lack academic rigours. So, to which extent this will impact the conceptual model. I think listing this important issue as
--

	a limitation of the review is not enough and there is a need to further discuss this in the discussion.
--	---

REVIEWER	Michael Schneider, PhD, DC University of Pittsburgh Pittsburgh, PA USA
REVIEW RETURNED	08-May-2020

GENERAL COMMENTS	This was a well-written and organized systematic review of the qualitative literature regarding the barriers/facilitators to the uptake and utilisation of digital interventions for self-management of low back pain.
--

REVIEWER	Silvano Mior Canadian Memorial Chiropractic College Canada
REVIEW RETURNED	09-Aug-2020

GENERAL COMMENTS	In this timely and informative literature synthesis, the authors critically appraise qualitative studies assessing the role of digital health interventions to support self-management of low back pain (LBP). The authors provide a clear description of their methods, adequately summarize their results using an appropriate theoretical framework, and apply findings for future consideration in developing strategies for use of digital interventions in management of LBP based on a conceptual model. In general, this is a well written and conceptualized literature synthesis. Offer the following suggested comments for consideration: Line 84-88: Although defined in the inclusion criteria (Table 1), it would be helpful to readers if the authors briefly define DHIs in the introduction. This will provide appropriate context for the reader. Lines 174-175: Appreciating the differing opinions regarding exclusion of studies of questionable methodological quality, could the authors please include an explanation of why they decided a priori to include all critically appraised studies regardless of their quality? Line 240: Authors suggest Table 4 describes sampling procedures but such is not so clearly described. Did studies describe use of purposive sampling? Was such sampling augmented by snowball sampling? How were participants selected? Further clarity would be helpful to inform Table 4, appreciating of course that further information is presented in the Supplementary file. Line 247: Please include a brief summary of the items that were most frequently not reported in the include studies. Line 284: Do you mean, "Accessibility to a computer was surprisingly not a requirement...", rather than "Accessibility to a computer was not surprisingly a requirement..."? I suggest the former is intended wording. Lines 445 and 453: Suggest changing "...this review" to "...our review" to clarify authors are referring to their review. Line 445-446: Please clarify the point of reference for the statement "..., although this may be due to the small number of studies and participants." Do the number of studies and participants in the authors' study differ from those in other reviews? Please provide some context for the noted statement. Line 511-513: In consideration of the detail provided in the methods, what specific "constraints" may have been involved, and how could potential useful studies have been "inadvertently excluded"? Please expand further in the paper.
---

VERSION 1 – AUTHOR RESPONSE

Reviewer(s)' Comments to Author:

Reviewer: 1

We thank Dr Fadi Al Zoubi for his constructive comments.

1. The introduction for DHI was good. However, it is expected from the authors to define it since it involves diverse range of e-interventions.

Authors response: This is a useful point and we have now included a definition of what we mean by a DHI in our introduction, page 5, lines 101-103: *“health interventions accessed through a computer, mobile phone, or other handheld device, involving a web-based programme, desktop programme or application”*.

2. In your eligibility criteria, there is a limitation of population to include only non-specific LBP. However, the title does not reflect that, and the introduction of the manuscript does not provide any information about this type of LBP. In addition, in your search strategy you included general LBP without referring to non-specific.

Authors response: We have removed “non-specific” from the eligibility criteria.

3. I am not familiar with the term "information specialist". Does that mean this person is a librarian or not?

Authors response: Yes, this person is a librarian and we have edited the manuscript to use this term instead of “information specialist” so that it is clear to all readers.

4. The authors mentioned in the study selection line 155-156 that any disagreement between the two reviewers results in inclusion of the citation to the next screening level. What is the next screening level? is it a third reviewer who decide? voting between the 4 reviewers? or full-text screening? I think there is a need to be explicit about these steps.

Authors response: Title and abstract screening were conducted by 2 reviewers, if there was disagreement the citation automatically went to the next screening level, either abstract or full-text. However, at full-text level any disagreement was discussed and a consensus agreed between the two reviewers and a third member of the team who had not conducted full-text screening of the citation. This paragraph has now been updated in the manuscript, page 8, lines 175-178: *“Title and abstract screening were performed by two of four independent reviewers (JK, MaS, KC, KW) using Distiller SR. Any disagreement between the two reviewers at title screening level resulted in inclusion of the citation to abstract level and subsequently any disagreement at abstract level resulted in inclusion of the citation to the full-text screening level.”*

5. The numbers reported in PRISMA Chart are different from the ones on results lines 208-209. The full-text screening was 100 in the text and 105 in PRISMA. The excluded studies in the text were 7436 and PRISMA 8113. Please review them

Authors response: In the original manuscript the “100” referred to the number excluded at the full-text screening stage; 8113 was the number excluded at both title and abstract screening combined. This has been made clearer in the text of the updated manuscript, page 8, lines 231-233: *“8113 were excluded following title and abstract screening (7436 at title level and 677 at abstract level) and a further 100 citations were excluded after full text screening.”*

6. As the author mentioned in the text that many of the DHIs were commercial and lack academic rigours. So, to which extent this will impact the conceptual model. I think listing this important issue as a limitation of the review is not enough and there is a need to further discuss this in the discussion.

Authors response: This is a preliminary conceptual model specific to the uptake and utilisation of a DHI to support people to self-manage their LBP. Due to the lack of literature in this field, the model is limited to four studies to date. However, as more rigorous studies are conducted and reported this model should be further developed and amended. The paragraph on page 31, lines 560-566 of the updated manuscript now reads: *“Finally, due to the lack of literature in this field, our conceptual model for the uptake and utilisation of DHIs to support the self-management of LBP is limited to four studies. It is possible that not all the important barriers and facilitators may have been identified, and thus our conceptual model must be considered preliminary. As more rigorous studies are conducted and reported this model should be further developed and amended. This information will be of particular use to those involved in designing and implementing DHIs focused on self-management of LBP and more widely.”*

Reviewer: 2

We thank Dr Schneider for his positive review.

Reviewer: 3

We thank Silvano Mior for their constructive comments.

1. Line 84-88: Although defined in the inclusion criteria (Table 1), it would be helpful to readers if the authors briefly define DHIs in the introduction. This will provide appropriate context for the reader.

Authors response: We agree with this omission from the original introduction and as per our response to Reviewer 1’s comment 1, we have now included a definition of what we mean by a DHI in our introduction, page 5, lines 101-103: *“health interventions accessed through a computer, mobile phone, or other handheld device, involving a web-based programme, desktop programme or application”*.

2. Lines 174-175: Appreciating the differing opinions regarding exclusion of studies of questionable methodological quality, could the authors please include an explanation of why they decided a priori to include all critically appraised studies regardless of their quality?

Authors response: We did not exclude papers based on quality appraisal because our aim was to develop a complete catalogue of barriers and facilitators to uptake and utilisation of DHIs in the context of LBP and we were eager to avoid missing important ideas. Furthermore, there remains no clear agreement on how best to appraise qualitative research. The manuscript has been updated to include this view and reference on page 9, lines 196-197: “*A priori cut-off points were not determined as studies were not excluded on the basis of methodological quality due a lack of clear agreement on how best to apprise qualitative literature.*”

Reference: Dixon-Woods M, Sutton A, Shaw R, Miller T, Smith J, Young B, Bonas S, Booth A, Jones D (2007) Appraising qualitative research for inclusion in systematic reviews: a quantitative and qualitative comparison of three methods. *J Health Serv Res Policy* 12: 42–47.

3. Line 240: Authors suggest Table 4 describes sampling procedures but such is not so clearly described. Did studies describe use of purposive sampling? Was such sampling augmented by snowball sampling? How were participants selected? Further clarity would be helpful to inform Table 4, appreciating of course that further information is presented in the Supplementary file.

Authors response: Full details provided on sampling procedures provided in the published articles are included in Table 4, with some minor updates from the original submission. This has been acknowledged further in the updated manuscript by editing the text on page 18, lines 266-269: “*Sampling procedures used for the qualitative component of the included studies (Table 4) were described for three of the studies as an invitation to participants to take part in an interview varied across studies and for the fourth study where the qualitative component was part of a survey, all participants took part.*”

Only one study (OneSelf) described their procedure as a purposive and convenient sample; this is reported in Supplementary file 2.

4. Line 247: Please include a brief summary of the items that were most frequently not reported in the include studies.

Authors response: Supplementary file 2 details all quality appraisal items that were included in each study. However, the text has been updated to give further examples, page 18, lines 280-281: “*Items within domain 1 (Research team and reflexivity) generally had very poor reporting with several items not reported by any studies, for example researcher occupation and experience and training were not reported by any of the included studies.*” Further updates have been made to this paragraph in the manuscript to ensure that it correctly aligned with Supplementary File 2.

5. Line 284: Do you mean, “Accessibility to a computer was surprisingly not a requirement...”, rather than “Accessibility to a computer was not surprisingly a requirement...”? I suggest the former is intended wording.

Authors response: Thank you for pointing out this confusion, indeed the first wording given here is correct and the manuscript, page 21, line 315 has been updated accordingly.

6. Lines 445 and 453: Suggest changing “...this review” to “...our review” to clarify authors are referring to their review.

Authors response: This update has been made on page 27, line 464.

7. Line 445-446: Please clarify the point of reference for the statement “..., although this may be due to the small number of studies and participants.” Do the number of studies and participants in the authors’ study differ from those in other reviews? Please provide some context for the noted statement.

Authors response: The O’Connor et al and Hardiker & Grant reviews that we have referred to included 19 and 50 studies, respectively, compared to 4 studies in our review. The text on page 28, lines 477-478 has been updated to add context to this statement: *“although this may be due to the small number of studies and participants in our review compared to O’Connor et al and Hardiker & Grant, reviews which included 19 and 50 studies, respectively.”*

8. Line 511-513: In consideration of the detail provided in the methods, what specific “constraints” may have been involved, and how could potential useful studies have been “inadvertently excluded”? Please expand further in the paper.

Authors response: Our original wording of “Our search strategy involved a number of constraints and focussed on published literature which may have inadvertently excluded potentially useful studies.”, referred to our inclusion and exclusion criteria as “constraints”. However, on reflection this is not the most suitable wording. Our criteria were used as they were appropriate to answer our research question and to identify all published qualitative literature in this field; and rather given the nature of DHIs, we may have excluded literature that was not presented as a research study or published in scientific literature. We have updated the first paragraph on limitations of our study to incorporate Reviewer 3’s point (page 30/31, lines 542-551 of updated manuscript): *“Our review does however have some limitations. Many DHIs are developed commercially and do not undergo formal academic evaluation (15) resulting in relatively sparse literature in this area. Our search strategy involved several eligibility criteria, including that studies must be published in peer-reviewed academic journals, and as such we did not identify any grey literature. However, it is unlikely that such findings, if available, would have held scientific rigour and added to the findings of this review. Further, as our analysis and synthesis of data was based on reviewing published literature, not the original data, this could have impacted on the background context to some of the quotes used in this manuscript.”*

VERSION 2 – REVIEW

REVIEWER	Fadi Al Zoubi Hong Kong Polytechnic University, China
REVIEW RETURNED	17-Oct-2020

GENERAL COMMENTS	No more comments
------------------

REVIEWER	Silvano Mior Canadian Memorial Chiropractic College, Canada
REVIEW RETURNED	21-Oct-2020

GENERAL COMMENTS	This is a well written and conceptualized review. Several minor issues are noted that would benefit from further clarity. Line 52: In the results section of the abstract, suggest providing a brief summary of numbers of citations retrieved, underwent full text screening and final inclusion. Table 4, pg 14, line37, column 3: consider removing “in” from description of inclusion criteria. Line 254: Sampling procedures used in the studies was described in Table 4; however, there’s a discrepancy between numbers described in Table 3 and Table 4. For example, in the Oneself study, invitations were sent to 238 eligible participants but only 18 were involved in the qualitative study. A comment about sampling procedures is made in the “limitation” section (lines 538-540); however, please clarify the sampling strategy used by the authors to recruit participants (purposive, snowball, etc) as per COREQ item 10. The description of sampling procedures requires clarification in Table 4 and on lines 254 and 255. Line 261: Please clarify what approach (methodological orientation) studies used to analyze their data rather than simply note they identified themes? Different approaches may provide different coding strategies rendering variation in thematic analysis and sense if saturation was achieved. Lines 268-270: In regard to data saturation, the authors provide no comment if included studies addressed the issue of data saturation. It would be helpful if such is included.
---

VERSION 2 – AUTHOR RESPONSE

Response to reviewer’s comments

Reviewer 3 Comments to Author:

We thank Silvano Mior for their helpful comments to improve the clarity of the manuscript.

1. Line 52: In the results section of the abstract, suggest providing a brief summary of numbers of citations retrieved, underwent full text screening and final inclusion

Author Response – Thank you for this useful suggestion. Additional figures relating to the screening process (number of citations identified and number full text screened) have been added to the abstract at lines 52 and 53: “Our systematic search resulted in the identification of 14191 citations, of

which 105 full-text articles were screened, resulting in the inclusion of five full text articles from four studies.”

2. Table 4, pg 14, line37, column 3: consider removing “in” from description of inclusion criteria.

Author Response – We have removed “in” from this sentence.

3. Line 254: Sampling procedures used in the studies was described in Table 4; however, there’s a discrepancy between numbers described in Table 3 and Table 4. For example, in the Oneself study, invitations were sent to 238 eligible participants but only 18 were involved in the qualitative study.

Author Response – Thank you for highlighting this issue, we appreciate that the different numbers may cause some confusion. The 238 participants referred to in Table 4, were the number of participants who took part in the overall study, all of whom were further invited to participate in the qualitative component of the study. The 18 participants referred to in table 3 were the final number of participants who agreed to participate in the qualitative study.

In order to reduce confusion regarding the different numbers, we have added in an extra bullet point in Table 4 to show the total number of users invited and number who agreed to participate: “238 users invited to participate, 18 agreed.”

4. A comment about sampling procedures is made in the “limitation” section (lines 538-540); however, please clarify the sampling strategy used by the authors to recruit participants (purposive, snowball, etc) as per COREQ item 10. The description of sampling procedures requires clarification in Table 4 and on lines 254 and 255.

Author Response – In order to provide more clarity on this point we have made changes to Table 4 to include the sampling strategies utilised for the Oneself study and Get Well Fast study which were not previously included. Furthermore, we have added further details at lines 256-259 outlining the sampling strategies used in each study: “Several sampling strategies were utilised, including purposive (28, 29) and convenience sampling (28, 29, 37), while in another study participants were sampled consecutively (39). In the further study, where the qualitative component was part of a self-administered survey, all participants took part (38).”

5. Line 261: Please clarify what approach (methodological orientation) studies used to analyse their data rather than simply note they identified themes? Different approaches may provide different coding strategies rendering variation in thematic analysis and sense if saturation was achieved.

Author Response – We have provided additional information at lines 264-265 outlining the approach each of the studies used to analyse their data: “Data was then analysed inductively (28), using grounded theory (29), thematically (37, 38) and using content analysis (39) to identify common themes.”

6. Lines 268-270: In regard to data saturation, the authors provide no comment if included studies addressed the issue of data saturation. It would be helpful if such is included.

Author Response – We have made changes to incorporate this important point. Only one of the included articles referred to data saturation - a sentence has been added at lines 265-266 describing this: “Just one article (29) referred to data collection and analysis continuing until data saturation was achieved.”.